# New Synthesized Activating Transcription Factor 3 Inducer SW20.1 Suppresses Resistin-Induced Metabolic Syndrome

**DOI:** 10.3390/biomedicines11061509

**Published:** 2023-05-23

**Authors:** Tu T. Tran, Wei-Jan Huang, Heng Lin, Hsi-Hsien Chen

**Affiliations:** 1International Ph.D. Program in Medicine, College of Medicine, Taipei Medical University, Taipei 110, Taiwan; d142109017@tmu.edu.tw; 2Department of Internal Medicine, Thai Nguyen University of Medicine and Pharmacy, Thai Nguyen 241-17, Vietnam; 3School of Pharmacy, Taipei Medical University, Taipei 110, Taiwan; 4Department of Physiology, School of Medicine, College of Medicine, Taipei Medical University, Taipei 110, Taiwan; 5TMU Research Center of Urology and Kidney, Taipei Medical University, Taipei 110, Taiwan; 6Division of Nephrology, Department of Internal Medicine, School of Medicine, College of Medicine, Taipei Medical University, Taipei 110, Taiwan; 7Division of Nephrology, Department of Internal Medicine, Taipei Medical University Hospital, Taipei 110, Taiwan

**Keywords:** activating transcription factor 3, obesity, resistin, SW20.1

## Abstract

Obesity is an emerging concern globally with increasing prevalence. Obesity is associated with many diseases, such as cardiovascular disease, dyslipidemia, and cancer. Thus, effective new antiobesity drugs should be urgently developed. We synthesized SW20.1, a compound that induces activating transcription factor 3 (ATF3) expression. The results of Oil Red O staining and quantitative real-time polymerase chain reaction revealed that SW20.1 was more effective in reducing lipid accumulation in 3T3-L1 preadipocytes than the previously synthesized ST32db, and that it inhibited the expression of the genes involved in adipogenesis and lipogenesis. A chromatin immunoprecipitation assay indicated that SW20.1 inhibited adipogenesis and lipogenesis by binding to the upstream promoter region of resistin at two sites (−2861/−2854 and −241/−234). In mice, the intraperitoneal administration of SW20.1 reduced body weight, white adipocyte weight in different regions, serum cholesterol levels, adipogenesis-related gene expression, hepatic steatosis, and serum resistin levels. Overall, SW20.1 exerts antiobesity effects by inhibiting resistin through the ATF3 pathway. Our study results indicate that SW20.1 is a promising therapeutic drug for diet-induced obesity.

## 1. Introduction

Obesity is among the most prevalent metabolic disorders worldwide, affecting 61% of adults, and it is associated with diverse diseases, such as dyslipidemia, diabetes mellitus, hypertension, malignancy, and osteoarthritis [1,2]. These associations are caused by the excessive differentiation and proliferation of adipocytes. Adipogenesis is characterized by structural alterations, lipid accumulation, lipogenic enzyme production, and triglyceride (TG) deposition in adipocytes [3,4,5]. Thus, the inhibition of adipogenesis and TG deposition in adipocytes is a critical therapeutic target for drugs designed to treat or prevent obesity.

Adipose tissue consists of white adipose tissue (WAT), brown adipose tissue (BAT), and brite adipose tissue and is involved in both energy storage and thermogenesis regulation. Adipose tissue also secretes numerous hormones, cytokines, and metabolites, which are known as adipokines [6]. Resistin is an adipose tissue-derived signaling cysteine-rich protein. An elevated plasma resistin level is associated with various dysfunctions, including altered lipid and carbohydrate metabolism, inflammation, and insulin resistance [7]. Furthermore, resistin is related to knee osteoarthritis, ovarian cancer, and acute pancreatitis [8,9,10]. A human study revealed the association of resistin with obesity and impaired insulin sensitivity [11].

Lifestyle modification is the mainstay of obesity treatment [6]. However, when lifestyle modification is ineffective or obesity is severe, drug combination therapy is recommended. Although some antiobesity drugs have been approved and marketed, most have been discontinued because of their severe adverse effects [7,8]. Therefore, an effective and safe antiobesity drug should be urgently developed.

*Salvia miltiorrhiza* (or *S. miltiorrhiza* radix) is a member of the Labiatae family. Because of its beneficial ability to enhance cardiovascular health and reduce platelet aggregation, *S. miltiorrhiza* is a well-known herb used to treat numerous ailments in traditional Chinese medicine [12]. *S. miltiorrhiza* contains multiple chemical substances. More than 50 water-soluble components and 30 liposoluble components have been identified in *S. miltiorrhiza* [13]. Most water-soluble components of *S. miltiorrhiza* exhibit antioxidative, anticoagulant, and myocardial protective effects, which are mainly attributed to blood activation and blood clot elimination [14,15]. Primary tanshinones, which are liposoluble components of *S. miltiorrhiza,* exhibit antibacterial, anti-inflammatory, and myocardial protective effects and protective effects for endothelial cells [14,16]. In addition to tanshinone I and tanshinone IIA, several specific liposoluble components of *S. miltiorrhiza* have been determined to exert antiobesity effects [12,17,18].

Activating transcription factor 3 (ATF3), a member of the ATF/cAMP response element-binding (CREB) family, binds to the cyclic AMP response element in promoters containing the consensus sequence TGACGTCA [19,20]. ATF3 is a homodimer-forming transcriptional repressor. It forms a heterodimer with the members of the ATF/CERB or CCAAT/enhancer-binding protein (C/EBP) family to exert suppressive or stimulatory effects [21,22]. Ultraviolet radiation, cAMP, calcium influx, and cytokines are stress signals that can increase the generation of ATF3, which is normally produced at low levels in normal or quiescent cells [23,24]. ATF3 can inhibit the mRNA expression of peroxisome proliferator-activated receptor γ(PPARγ) and C/EBPα, thus suppressing adipocyte development [25,26]. High-fat diet (HFD)-fed ATF3^−/−^ mice gained more weight than did their littermate wild-type control mice and developed impaired glucose metabolism and hyperlipidemia [27,28]. Furthermore, 16C (an ATF3 stimulant) effectively alleviated metabolic syndrome in mice with HFD-induced obesity [29].

In our previous study, we identified *S. miltiorrhiza*-related chemically synthesized derivatives with high ATF3-inducing activity from the modified Chinese herb single-compound library at the National Research Institute of Chinese Medicine [27]. In addition to previously identified single compounds, including ST32da and ST32db, we discovered a new single compound, SW20.1 (a synthetic molecule similar to neo-tanshinlactone), with more lipid solubility. We hypothesized that SW20.1 exerts stronger antiobesity and anti-metabolic syndrome effects than does ST32db. Therefore, in this study, we investigated the antiobesity and anti-metabolic syndrome effects of SW20.1 and ST32db in mice with HFD-induced obesity and explored whether the molecular mechanism of SW20.1 involves the ATF3-medicated signaling pathway.

## 2. Materials and Methods

### 2.1. Cell Culture and Adipocyte Differentiation

We cultured 3T3-L1 mouse preadipocytes obtained from the American Type Culture Collection (Manassas, VA, USA) in Dulbecco’s modified Eagle’s medium (DMEM) supplemented with 10% (*v*/*v*) fetal bovine serum (FBS) and 100 U/mL penicillin/streptomycin (0.1 mg/mL). 3T3-L1 cells that reached confluence were cultured in differentiation medium consisting of DMEM, 0.5 mM IBMX, 10% FBS, 5 g/mL insulin, and 1 M dexamethasone. SW20.1 was dissolved in dimethyl sulfoxide (DMSO) to obtain a stock concentration of 25 nM. To determine the effects of SW20.1, 3T3-L1 cells were treated with SW20.1 for 8 days during cell differentiation. DMSO at the final concentration of 0.5% was used as the control.

### 2.2. Oil Red O Staining

After 8-day SW20.1 treatment during 3T3-L1 preadipocyte differentiation, the differentiated adipocytes were washed twice with phosphate-buffered saline and fixed for 1 h with 10% formalin. Then, the cells were stained with Oil Red O working solution for 30 min. Subsequently, the cells were washed four times with distilled water before optical microscopy analysis. The Oil Red O staining solution was eluted with 100% isopropanol (*v*/*v*), and its optical absorbance at 500 nm was measured.

### 2.3. Quantitative Real-Time Polymerase Chain Reaction

Total RNA was extracted from the cultured cells or adipose tissues by using the Trizol reagent (Invitrogen, Waltham, MA, USA). RNA was reverse transcribed to cDNA by using the iScript cDNA synthesis kit (Biorad, Hercules, CA, USA). Quantitative real-time polymerase chain reaction (qRT-PCR) was performed using a real-time PCR system (ABI StepOnePlus, Applied Biosystems, Grand Island, NY, USA) with SYBR green (Biorad). The primer sequences used for qPCR are listed in Table 1.

### 2.4. Chromatin Immunoprecipitation Assay

By using a magnetic ChIP kit (Pierce Magnetic ChIP Kit), a chromatin immunoprecipitation (ChIP) assay was performed on 3T3-L1 cells fixed with 1% formaldehyde (Thermo Fisher Scientific, Waltham, MA, USA). Chromatin was immunoprecipitated using an anti-ATF3 antibody (Abcam ab254268, Cambridge, UK). DNA was detected using standard PCR. The primers used for PCR are listed in Table 1.

### 2.5. Animal Studies

In this study, we used male C57BL/6 mice. To investigate the effects of SW20.1 and ST32db on obesity and metabolic syndrome, 7-week-old mice were fed an HFD (45% kcal from fat) with or without the intraperitoneal administration of SW20.1 (60 mg/kg/week) or ST32db (75 mg/kg/week) for 10 weeks, three times per week. In the control group, we injected the mice with 100% DMSO (1 mL/kg). The body weight and food intake of the mice were measured weekly throughout the experiments. After treatment, insulin sensitivity and glucose tolerance were measured. The mice were then sacrificed, and their blood and tissue samples were collected. All procedures were performed in accordance with the guidelines of the Institutional Animal Care and Use Committee of Taipei Medical University, Taiwan.

### 2.6. Glucose and Insulin Tolerance Tests

For the glucose tolerance test (GTT), the mice were fasted for 16 h before being administered an intraperitoneal injection of glucose (2 g/kg body weight) in saline. Plasma glucose levels were measured from blood samples collected from the tail vein at 0, 15, 30, 60, 90, and 120 min after glucose injection. For the insulin tolerance test (ITT), the mice were fasted for 6 h before being administered an intraperitoneal injection of 1 U/kg body weight of insulin in saline. Plasma glucose levels were measured from blood samples collected from the tail vein at 0, 15, 30, 45, 60, 90, and 120 min after insulin injection. We used a commercial glucose meter to measure plasma glucose levels [29].

### 2.7. Histology, Adipocyte Size Measurement, and Adipocyte Number Estimation

The samples were dissected and fixed with 4% paraformaldehyde overnight at 4 °C and then fixed in paraffin before sectioning and Hematoxylin and Eosin staining. To measure the adipocyte diameter and size, stained sections of inguinal white adipose tissue (iWAT) were analyzed using the Adiposoft app (version 1.16), an advanced distribution of ImageJ software created at the Imaging Unit of the Center for Applied Medical Research, University of Navarra, Pamplona, Spain [30].

### 2.8. Measurement of Biochemical Parameters

Blood samples (500 μL) were collected from the tail vein and centrifuged at 6000× *g* for 3 min to separate the serum from the cells. Within 24 h of blood collection, serum biochemical parameters were analyzed. The serum levels of blood urea nitrogen, creatinine, triglyceride, glutamic oxaloacetic transaminase, and glutamate pyruvate transaminase were determined using an automated analyzer (Spotchem EZ SP 4430, Azkray, Kyoto, Japan) [27].

### 2.9. Measurement of the Serum Resistin Level

Serum resistin level was measured using a mouse resistin enzyme-linked immunosorbent assay (ELISA) kit (Abcam, cat #ab205574) as per manufacturer’s instructions [27].

### 2.10. Statistical Analysis

All data are expressed as the mean ± standard error from at least three experiments. Statistical analysis was conducted using an unpaired *t*-test, one-way analysis of variance (ANOVA), and repeated two-way ANOVA followed by the Tukey test. A *p*-value of < 0.05 indicated significance.

## 3. Results

### 3.1. SW20.1 Induces ATF3 Expression

To determine whether SW20.1 is involved in inducing ATF3 expression, we compared the ability of each of the compounds (at a concentration of 30 µM; Figure 1b) to induce ATF3 expression. SW20.1 more strongly induced ATF3 expression than did the other derivates, especially ST32da and the control. We conducted a 2-day time-dependent experiment by using 30 µM SW20-1. The ATF3 expression induced by SW20.1 after a 3 h treatment peaked at 6 h and then declined (Figure 1c). In a dose-dependent experiment, the ATF3 expression induced by 15, 30, 60, and 120 µM SW20.1 significantly differed from that induced by the control (Figure 1c). Our findings indicate that the structure of this newly discovered ATF3 inducer significantly differs from that of the previously synthesized ST32da and ST32db. Moreover, SW20.1 more efficiently induced ATF3 expression than did ST32da and ST32db.

### 3.2. SW20.1 Suppresses 3T3-L1 Preadipocyte Differentiation through the Resistin Pathway

To determine whether the antiobesity effect of SW20.1 is stronger than that of the previously synthesized single compounds, we compared oil deposition in 3T3-L1 preadipocytes between SW20.1 and ST32db. As illustrated in Figure 2a,b, compared with 30 µM ST32db, 30 µM SW20.1 more strongly inhibited oil deposition. Compared with the control, SW20.1 more strongly suppressed the expression of the genes involved in adipogenesis and lipogenesis (Figure 2c,d). Furthermore, SW20.1 also induced a significant difference in the PPARα gene expression in the beta-oxidation process compared with the control (Figure 2e).

### 3.3. SW20.1 Inhibits Adipogenesis Directly through the ATF3-Resistin Pathway

Resistin gene expression is induced during adipocyte differentiation [31]. A clinical cohort study revealed that higher resistin levels are associated with higher adiposity [11]. In this study, we determined whether resistin is involved in the inhibition of adipocyte differentiation by SW20.1. Analysis of the proximal promoter regions of resistin performed using the PROMO 8.3 web platform (A virtual laboratory for the study of transcription factor binding sites in DNA sequences) revealed four potential ATF3 binding sites in the resistin promoter (Figure 3c). We determined the direct relationship between ATF3 and resistin after treatment with SW20.1. First, we transfected shRNA-ATF3 into 3T3-L1 adipocytes to inhibit ATF3 expression after SW20.1 treatment. As presented in Figure 3a, shRNA-ATF3 suppressed ATF3 expression, and resistin expression increased after ATF3 knockdown using shRNA-ATF3 (Figure 3b). These results suggest that after SW20.1 treatment, ATF3 directly regulates resistin expression. We conducted a ChIP assay with qRT-PCR to examine whether ATF3 binds to the proximal promoter regions of resistin at its potential binding sites. As presented in Figure 3d,e, ATF3 bound to sites 1 (−2861/−2854) and 4 (−241/−234) but not to site 2 (−2716/−2709) or 3 (−930/−923). Thus, SW20.1 suppresses adipogenesis/lipogenesis and adipocyte transdifferentiation through the ATF3-dependent pathway, directly inhibiting the resistin pathway.

### 3.4. SW20.1 Ameliorates Obesity and Obesity-Related Metabolic Syndrome in Mice

We examined whether SW20.1 modulates body weight and the development of obesity-induced disorder in the HFD-fed mice. After consuming an HFD for 1 week, the obese mice received an intraperitoneal injection of ST32db or SW20.1 at 75 or 60 mg/kg/week separately or a vehicle (Figure 4a). Their food consumption and body weight were recorded weekly. Both the SW20.1- and ST32db-treated mice exhibited lower body weight than did the untreated HFD-fed mice. However, food intake did not significantly differ between the placebo and the two treatment groups (Figure 4b). The weights of inguinal WAT (iWAT), epididymal WAT, mesenteric WAT, and retroperitoneal WAT were lower in the SW20.1- and ST32db-treated mice after 10 weeks of treatment than in the untreated mice. However, the weights of the interscapular BAT, epididymal BAT, and perivascular adipose tissue did not differ between the placebo and two treatment groups (Figure 4d). In addition, SW20.1 treatment enhanced glucose tolerance and reduced serum cholesterol levels in the HFD-fed obese mice (Figure 4e,g). We identified that SW20.1 also ameliorated hepatic steatosis (Figure 4i) and reduced the size and diameter of adipocytes in iWAT (Figure 4j,k).

### 3.5. SW20.1 Treatment Reduces Serum Resistin Levels to Inhibit Adipogenesis in HFD-Fed Mice

Our in vitro experimental results revealed that SW20.1 inhibited the expression of the resistin gene through the ATF3 pathway. Thus, we examined whether the reduction in body weight and white adipocyte weight in HFD mice was due to a decrease in resistin expression after SW20.1 treatment. ELISA results revealed lower serum resistin levels after SW20.1 and ST32db treatment (Figure 5a). In the analysis of the expression of genes involved in adipogenesis and lipogenesis in iWAT, we detected changes in the expression of genes involved in adipogenesis (Figure 5b,c). These results suggest that SW20.1 exerts antiobesity effects mainly by inhibiting adipogenesis through the resistin pathway.

## 4. Discussion

In this study, we discovered a new compound, SW20.1, that possessed the ability to induce ATF3 and exerted antiobesity effects. In addition, SW20.1 decreased cholesterol levels more effectively than did the previously synthesized single compound ST32db.

In our previous studies, we synthesized various single compounds that can induce ATF3 expression. In this study, we referred to the basic structures of ST32da and ST32db for modification. A series of chemically synthesized derivatives called SWs are presented in Appendix A. By using an ATF3-specific promoter screening approach based on the detection of luciferase activity in stable adipocyte clones [27], we determined that SW20.1 exhibited ATF3-inducing activity. Compared with other derivatives, including ST32db, SW20.1 exhibited superior lipid solubility (Figure 1a). SW20.1 and ST32db have a five-member ring condensed at positions 3 and 4. ST32db is a derivative of neotanshinlactone with modifications including saturation at the double bond on the five-member ring and the change in the position of the methyl group from the aromatic ring to the five-member ring, whereas SW20.1 has an additional benzyl group at carbon 8 on a coumarin scaffold. Therefore, SW20.1 penetrates cells more easily than do other derivatives. This mechanism may explain the more effective inhibition of oil deposition by SW20.1 in vitro in this study. By examining the beta-oxidation genes, we discovered that SW20.1 enhanced fatty acid oxidation via PPARα (a ligand-activated transcriptional factor that modulates the expression of genes involved in fatty acid beta-oxidation and is a key regulator of energy balance) [32]. However, the body weight reduction effect did not differ between ST32db and SW20.1. This finding may be due to the pharmacokinetic properties of SW20.1 are more effective than those of ST32db. Future studies are warranted to identify the reasons for this finding. In an in vitro experiment, we examined the cell cytotoxicity of SW20.1 by using an MTT assay with several concentrations of SW20.1 and verified its lipid accumulation effects. The results demonstrated that 30 μM SW20.1 exerted an effect similar to that of 40 μM SW20.1 and no cell death was detected during subsequent cell differentiation (Appendix A). Subsequently, we conducted dose- and time-dependent analyses of SW20.1 in 3T3-L1 cells. Our results revealed that the ATF3 expression induced by SW20.1 peaked at 6 h and then declined (Figure 1c), indicating that ATF3 undergoes ubiquitination under normal conditions. Future studies should examine whether SW20.1 binds to certain receptors after entry into cells to induce ATF3 gene and protein expression or whether SW20.1 inhibits certain proteins, thereby inhibiting the ubiquitination of ATF3 and finally increasing ATF3 levels in cells. Given the ubiquitination of ATF3, SW20.1 would have to be administered more frequently than are other drugs to achieve its antiobesity properties in clinical practice in the future.

Adipose tissues secrete a lot of adipokines, such as leptin, resistin, and adiponectin which is closely related to type 2 diabetes because of its effect on insulin sensitivity and inflammation [33]. Leptin inhibits appetite, stimulates thermogenesis, increases fatty acid oxidation, reduces glucose levels, and decreases body fat and weight [34]. However, mice with HFD-induced obesity exhibited leptin resistance [35], demonstrating the vicious pathology linking obesity and metabolic syndrome [36]. SW20.1 administration inhibited leptin gene expression. These results are similar to those for derivatives such as ST32da and ST32db [27,37]. Adiponectin is considered anti-inflammation and is positively associated with insulin sensitivity [33]. However, adiponectin also promotes adipocyte differentiation and lipid accumulation [38]. Our results validated the anti-adipogenesis effects of SW20.1, which are exerted through the inhibition of adiponectin, resistin, and leptin. The inhibition of adiponectin explains a part of the ITT results in the mice model in this study.

Our previous study indicated that the ST32da suppresses the ChREBP/SCD1 pathway, resulting in the reduction of lipogenesis/adipogenesis and the adipocyte browning process through ATF3 expression induction [27]. However, in our current study, we determined that SW20.1 suppressed the expression of genes related to adipogenesis and lipogenesis, but not those involved in the ChREBP/SCD1 pathway (Figure 5b,c) and the adipocyte browning process; this result indicates that SW20.1 may induce ATF3 expression independently of the ChREBP/SCD1 pathway; that is, SW20.1 downregulates adipogenesis-/lipogenesis-related gene expression through the inhibition of resistin. Resistin was identified to mediate the relationship between obesity and insulin resistance [7]. Elevated resistin levels exacerbate mitochondrial damage and hepatic steatosis through the AMP-activated protein kinase (AMPK)/PPARγ coactivator 1α signaling pathway [39]. Young-An Bae et al. demonstrated that ATF3 induction occurred downstream of AMPK activation in RAW264.7 murine macrophages [40]. These findings suggest that both ATF3-dependent and -independent pathways are involved in the antiobesity effects of SW20.1. Future studies should use ATF3-knockout mice treated with SW20.1 to elucidate the mechanisms underlying these findings.

We conducted a trial to identify the suitable dose of SW20.1 for this experiment. At the ages of 8 to 10 weeks, the mice in the control and SW20.1 groups were nervous and played with their food, and substantial amounts of food were identified in their bedding. Therefore, the difference in food intake between the control and SW20.1 groups and the ST32db group (Figure 4b) did not result from food intake. In addition to reducing the body weight, SW20.1 reduced the length of mice compared with the control mice (Figure 4c). This finding indicated that SW20.1 possibly decreased the muscle mass of the mice and inhibited their bone development. To verify this hypothesis, certain markers, such as osteocalcin and amino-terminal propeptide of procollagen I, should be examined [41]. However, due to the limitation of our study, we did not investigate these markers, and intend to do so in future research. SW20.1 exhibited greater effectiveness than did ST32db in the GTT (Figure 4e). In our study, the administration of the intraperitoneal injections of SW20.1 or ST32db was started after 1 week of acclimation, and the treatment duration was 10 weeks. By contrast, in a previous study [37], the authors injected ST32db for 12 weeks after 4 weeks of acclimation. Thus, we speculate that SW20.1 exerted effects on glucose tolerance earlier than ST32db did. Glucose is transported into cells through facilitated diffusion and carrier-mediated mechanisms. Glucose transporters and insulin play critical roles in insulin-sensitive tissues, particularly the heart, skeletal muscle, and adipose cells. Other factors also affect glucose transport, including epinephrine and vigorous exercise [42]. This observation explains why the effect of SW20.1 in the ITT did not differ from that of the placebo (Figure 4f). However, the effect of SW20.1 in the GTT differed from that of placebo only after 15 min. These findings indicate that SW20.1 affects other processes or proteins facilitating glucose uptake in cells. Compared with the control, SW20.1 significantly reduced cholesterol levels (Figure 4g). However, the effect of ST32db on the reduction of cholesterol levels did not differ from that of placebo. SW20.1 administration did not reduce TG levels more than did the placebo. Catherine et al. described that an HFD did not alter plasma TG levels. By contrast, they reported that an HFD enhanced total plasma cholesterol levels from 16 weeks to 24 weeks of age [43]. Thus, in this study, we observed that SW20.1 administration reduced the cholesterol level. The toxic effects of SW20.1 on the liver and kidney did not differ from those of placebo (Appendix A).

Resistin mediated the relationship between obesity and insulin resistance because its expression in adipose tissue is proportional to fat size [7]. Another study suggested that resistin knockdown suppresses lipid production and activates fatty acid β-oxidation [31]. We postulate that SW20.1 regulates resistin expression through the induction of ATF3. In this study, we identified four potential binding sites of ATF3 in the resistin promoter. However, we determined that ATF3 binds to the resistin promoter at only two sites, specifically site 1 (−2861/−2854) and 4 (−241/−234). ATF3 is an unusual member of the ATF/CREB family of bZIP transcription factors in that it can both repress and activate transcription [44] by binding to different proteins. For instance, the transcription factor Jun associated with ATF3 plays a role in activating transcription, whereas histone deacetylase 1 associated with ATF3 represses transcriptional activity [45,46]. Chung et al. demonstrated that stimulatory protein 1 (Sp1) binds to the resistin promoter region in response to adipocyte differentiation [47]. In addition, ATF3 interacts with Sp1 through the GC box region of the DINE gene in injured neurons [48]. Future studies should determine the association between Sp1 and ATF3.

This study has some limitations that should be addressed. Currently, four mouse strains have been used to establish an HFD-induced obesity model: Kunming, C57BL/6, BALB/c, and ICR. ICR is considered to be the most suitable for testing drugs [49]. Humans consume not only HFDs but also other foods that contribute to obesity. Therefore, our model is only applicable to a subset of the obesity population. The use of leptin-deficient ob/ob mice can address the drawbacks of our experimental design [50]. Treatments must be developed for obesity and obesity-related metabolic syndrome. ATF3 may be a target in obesity treatment. The results of this study indicate that SW20.1 may prevent obesity and its related metabolic syndrome by inducing ATF3 expression and inhibiting the crucial upstream resistin-related pathway. SW20.1 is more effective for the prevention of diabetes than is the previously synthesized drug ST32db. ST32da plays a role in the browning of white adipocytes [27]. Therefore, from a clinical perspective, the combined use of SW20.1 and ST32da may be ideal for reducing and preventing obesity.

## 5. Conclusions

Our study indicated the effectiveness of SW20.1, an ATF3 expression inducer, in reducing and preventing obesity and its related metabolic disorder through the ATF3–resistin pathway. From a therapeutic perspective, SW20.1 is a promising drug for obesity treatment. Future studies should focus on the transcriptional and epigenetic regulation of ATF3 in adipogenesis.

## Figures and Tables

**Figure 1 biomedicines-11-01509-f001:**
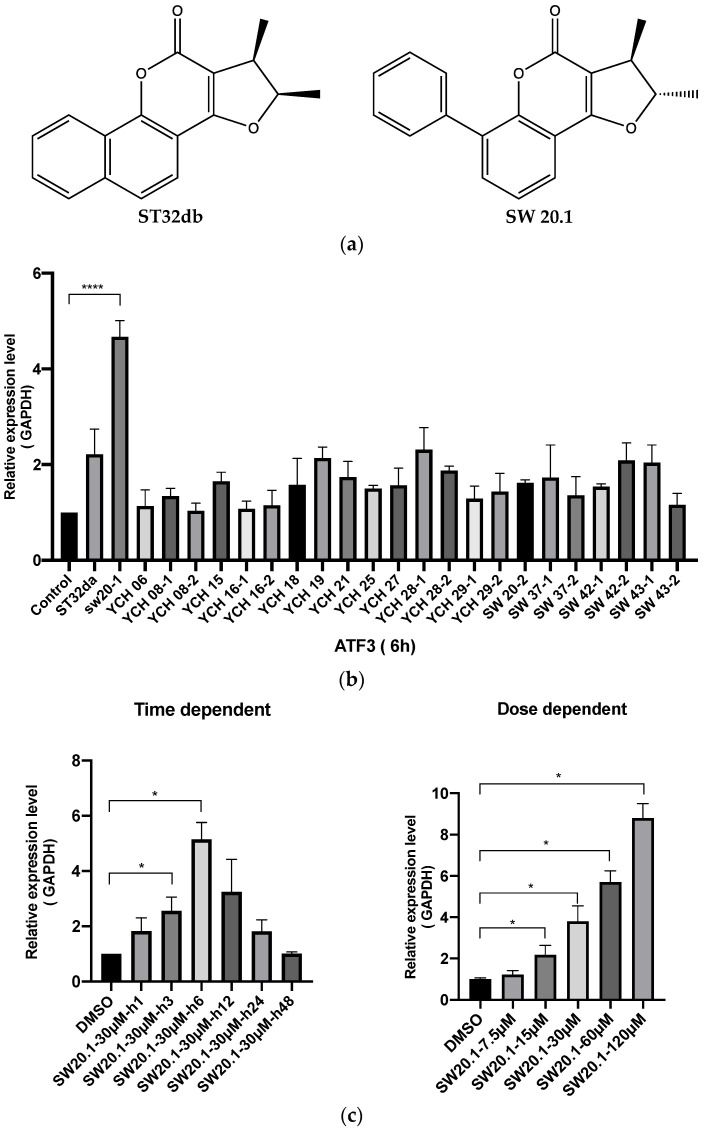
Characterization of the activating transcription factor 3 (ATF3) expression ability of SW20.1 in preadipocytes (3T3-L1 cells). (**a**) Chemical structure of SW20.1 and ST32db. (**b**) ATF3 gene expression induced by SW20.1 and other derivatives assayed through the quantitative real-time polymerase chain reaction (qRT-PCR). (**c**) Time- and dose-dependent mRNA ATF3 expression measured using qRT-PCR. Statistical comparisons were performed using unpaired *t*-test (**b**,**c**). Data are presented as mean ± standard error; * *p* < 0.05 and **** *p* < 0.0001 compared with the control.

**Figure 2 biomedicines-11-01509-f002:**
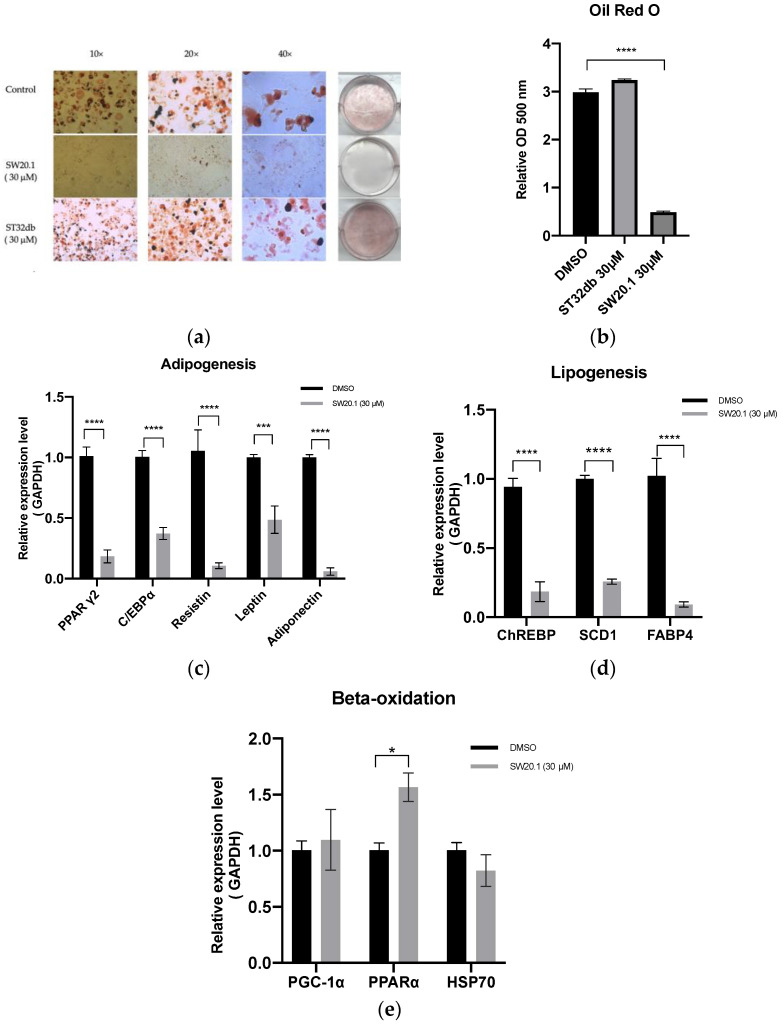
SW20.1 suppresses lipid accumulation and adipogenesis/lipogenesis. (**a**) Oil Red O staining of differentiated 3T3-L1 adipocytes treated with dimethyl sulfoxide, SW20.1, or ST32db for 8 days. (**b**) Relative qualification of the lipid content. (**c**) Analysis of expression of genes involved in adipogenesis, such as genes encoding leptin, resistin, peroxisome proliferator-activated receptor γ2 (PPARγ2), CCAAT/enhancer-binding protein α (C/EBPα), and adiponectin. (**d**) Analysis of the expression of genes involved in lipogenesis, such as the genes encoding carbohydrate-responsive element binding protein (ChREBP), stearoyl-CoA 9-desaturase (SCD1), and fatty acid binding protein 4 (FABP4). (**e**) Analysis of the expression of genes involved in beta-oxidation, such as the genes encoding peroxisome-proliferator-activated receptor-γ coactivator- 1alpha (PGC1α), PPARα, and heat shock protein 70 (HSP70). Statistical comparisons were performed using one-way ANOVA (**b**) and unpaired *t*-test (**c**–**e**). Data are presented as mean ± standard error; * *p* < 0.05; *** *p* < 0.001, and **** *p* < 0.0001.

**Figure 3 biomedicines-11-01509-f003:**
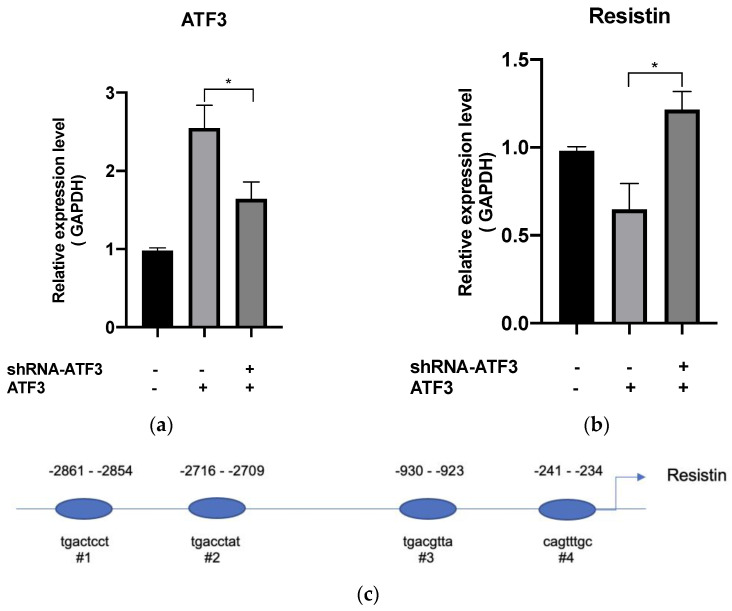
Molecular mechanism of SW20.1. (**a**) Activating transcription factor 3 (ATF3) expression was reduced by shRNA-ATF3. (**b**) Resistin expression was recovered after the knockdown of ATF3 through shRNA-ATF3. (**c**) Sequences of four potential binding sites for ATF3 in resistin promoter, including sites 1 (−2861/−2854), 2 (−2716/−2709), 3 (−930/−923), and 4 (−241/−234). (**d**,**e**) Results of chromatin immunoprecipitation experiments with ATF3-specific antibody and primers to amplify the sites 1 and 4 of the resistin locus, which contained one predicted ATF3 binding site in 3T3-L1 preadipocytes. Statistical comparisons were performed using an unpaired *t*-test (**a**,**b**,**d**,**e**). Data are presented as mean ± standard error; * *p* < 0.05; ** *p* < 0.01.

**Figure 4 biomedicines-11-01509-f004:**
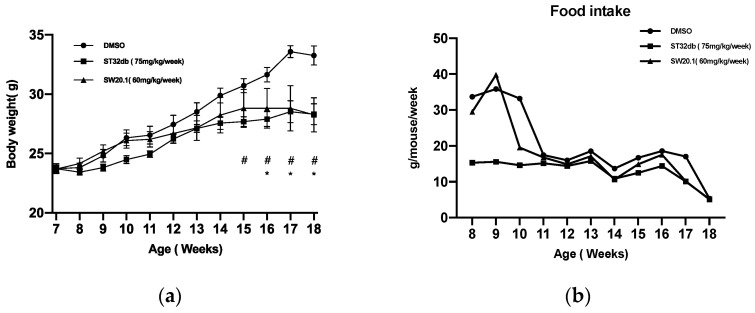
Results of treatment with or without SW20.1 or ST32db in wild-type mice with high-fat-diet (HFD)-induced obesity and metabolic syndrome. (**a**–**c**) Body weight, food intake, and body imaging changes of mice with HFD-induced obesity during 10 weeks of treatment with SW20.1 or ST32db (three times per week). (**d**) Weights of inguinal white adipose tissue (WAT), retroperitoneal WAT, mesenteric WAT, epididymal WAT, interscapular brown adipose tissue (BAT), epididymal BAT, and perivascular adipose tissue. (**e**) Glucose tolerance test results. (**f**) Insulin tolerance test results. (**g**,**h**) Cholesterol and triglyceride serum levels. (**i**) Inguinal WAT and liver hematoxylin and eosin staining results (scale bar = 100 μm). (**j**,**k**) Size and diameter of adipocytes in each group. Statistical comparison between the three groups were performed using repeated two-way analysis of variance (ANOVA) (**a**), one-way ANOVA (**d**,**g**,**h**,**j**,**k**), and unpaired *t*-test to compare pairwise groups at different time points (**e**,**f**). Data are presented as the mean ± standard error; ns, non-significant; * *p* < 0.05, ** *p* < 0.01, and **** *p* < 0.0001 for SW20.1 compared with the control and # *p* < 0.05 for SW20.1 compared with ST32db (control and ST32db group, *n* = 5; SW20.1 group, *n* = 4).

**Figure 5 biomedicines-11-01509-f005:**
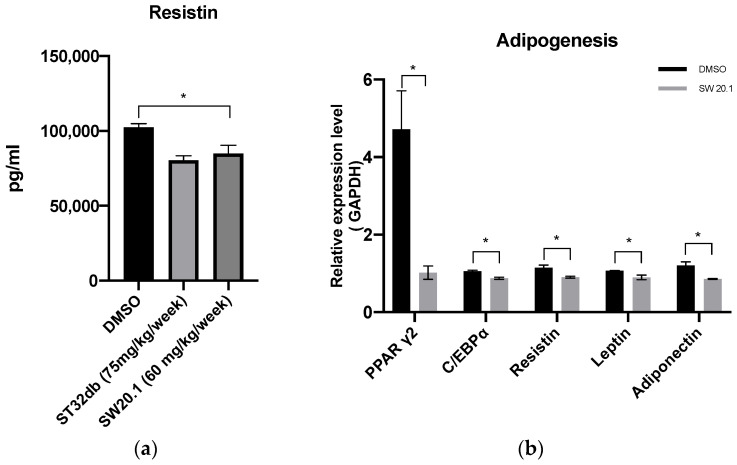
SW20.1 suppresses adipogenesis in high-fat-diet (HFD)-fed mice. (**a**) Serum protein levels of resistin determined using enzyme-linked immunosorbent assay in HFD-fed mice. (**b**) Expression of genes involved in adipogenesis in the inguinal white adipose tissue (iWAT). (**c**) Expression of lipogenesis genes in iWAT. Statistical comparisons were performed using one-way analysis of variance (**a**) and unpaired *t*-test (**b**,**c**). Data are expressed as mean ± standard error; ns, non-significant; * *p* < 0.05 (*n* = 3 per group).

**Table 1 biomedicines-11-01509-t001:** Primer sequences used in study.

Gene	Forward Primer	Reverse Primer
*ATF3*	CTCCTGGGTCACTGGTATTTG	CCGATGGCAGAGGTGTTTAT
*C/EBPα*	GTAACCTTGTGCCTTGGATACT	GGAAGCAGGAATCCTCCAAATA
*PPARγ2*	CTGGCCTCCCTGATGAATAAA	AGGCTCCATAAAGTCACCAAAG
*Adiponectin*	AAGGGCTCAGGATGCTACTGTT	AGTAACGTCATCTTCGGCATGA
*Resistin*	CTAAGCTGAGGGTCTGGAAATG	CACACACCCTTCTCCACTAAAG
*Leptin*	GGTTGATCTCACAATGCGTTTC	TGGGAGACAGGGTTCTACTT
*ChREBP*	TGTTCAGCATCCTCATCCGACCTT	TGAGTTGGCGAAGGGAATTCAGGA
*SCD1*	TGGGTTGGCTGCTTGTG	GCGTGGGCAGGATGAAG
*FABP4*	GCTCCTCCTCGAAGGTTTAC	CCCACTCCCACTTCTTTCAT
*GAPDH*	GGAGCCAAACGGGTCATCATCTC	GAGGGGCCATCCACAGTCTTCT
*HSP70*	TGGTGCTGACGAAGATGAAG	CGCTGAGAGTCGTTGAAGTAG
*PGC1α*	CTAGCCATGGATGGCCTATTT	GTCTCGACACGGAGAGTTAAAG
*PPARα*	ACCACTACGGAGTTCACGCATG	GAATCTTGCAGCTCCGATCACAC
Primer 1	ATCTGTTTATCTGCTGGTTCCAT	ACATGCACATGTGCACGTGTGT
Primer 2	TGCTTTAATTCCTTTGCTGTGT	AGAGCTAGCAAGATGGTTTGCTG
Primer 3	TAATCTCAATTTTGTCCTATC	AAACATTATTTAGTCATTATTGC
Primer 4	GCAAGGGAGCAGTTGACTAGAT	GGACGTTGTCTGGAGATAACACC

## Data Availability

The data presented in this study are available in the article and the Appendix A.

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
