# Peer review of "New Synthesized Activating Transcription Factor 3 Inducer SW20.1 Suppresses Resistin-Induced Metabolic Syndrome"

_biomedicines, 2023, doi:10.3390/biomedicines11061509_

Round 1
Reviewer 1 Report
Biomedicines – 2348195
General comments:
Tran and colleagues describe the efficacy of a newly synthesized anti-obesity compound using both in vitro and in vivo modeling. In short, they report that the compound has a significant effect on ATF3 expression in vitro and is associated with a reduction in body mass in high fat fed B6 mice in vivo. These effects may be mediated downstream of reduced resistin signaling activity and its subsequent effects on adipogenesis. Metabolic function is somewhat improved, although not as significantly as the authors indicated, especially in the context of insulin sensitivity/glucose tolerance.
Specific comments (in no order of importance):
There are typos here and there in the text. Please edit for clarity.
I would recommend changing the order of the last two paragraphs of the introduction. Specifically, discuss medicinal plants first, then ATF3. Otherwise, it is unclear why ATF3 is introduced in the introduction as it is currently written.
Perhaps I missed it, but what is the final concentration of DMSO used for each experiment?
Why were only male mice used?
Daily injections for 10 weeks are very stressful. It would be interesting to see how the response may have differed in unhandled mice relative to the DMSO injected mice to account for this potential confound.
How were the doses for the in vivo experiments chosen?
Figure 1 is unnecessary as is. It is recommended that the structure of only the prior compounds of interest (ST32da/b) and the compound studied here be included. This is also suggested for the Figure 2a. There is a lot of data provided that makes no meaningful contribution to the ms.
I understand that 30 um had a significant effect on ATF3 expression, but it was not maximal. Why was this concentration rather than a higher one used? Ae the higher concentrations toxic?
Figure 3 is too “busy.” I would recommend breaking it into two different figures.
From what is provided in the methods section, a repeated measures ANOVA was not used when appropriate. For example, when comparing the body weight data, food intake data, as well as the GTT and ITT data among the groups. Highlighting two individual timepoints over the course of the GTT does not demonstrate improved glucose tolerance, especially since these data were not analyzed correctly. Moreover, the ITT data suggests that both compounds are actually worse than DMSO alone.
There is clearly a difference in the length of the treated versus untreated mice. Is it statistically different? This is important for two reasons. First, it suggests that these compounds can alter linear (i.e., skeletal) growth and not just adipose tissue. It also raises the possibility that skeletal mass and other organs such as skeletal muscle could be affected. Taken together, this could result in the significant difference in total body mass among the treatment groups. Second, the mass of the individual adipose depots has not been adjusted for the difference in body length among the groups. In some cases it may not matter, but for other depots the significant difference may no longer be observed if the relative mass is used.
It is surprising that the degree of food intake between ages 8 – 10 weeks was not significantly different among the groups using the statistical model that was described. It is more than a 100% difference at times.
The discussion makes a number of unsubstantiated claims concerning the mechanistic basis for the observed effects. For example, differential degrees of ubiquitination and subsequent turnover of ATF3 is invoked, but there are no data. Similarly, there are not data indicating that SW20.1 is better able to cross the plasma membrane, there is nothing about glucose transporter expression, et cetera. If these data are available, then they must be included. Otherwise, these statements need to be more speculative.
It is not clear why leptin (and none of the other molecules included in that panel) are highlighted. In particular, an elevated level of adiponectin is beneficial. This contrasts with what is reported in Figure 3c, at least at the mRNA level. How is this reconciled?
.
Author Response
Dear Reviewer!
Please see the attachment

Reviewer 2 Report
Reviewers’ Comments (Remarks to the Authors):
The authors address one of the main global health problems, with a high mortality rate and highly related to multiple pathologies, obesity. In this manuscript, the authors establish and describe a new family of molecules, likely to be a guaranteed therapy in diet-induced obesity models. In a detailed way, the authors encourage a reading by the reader, making the paper with a great scientific value to the MDPI group
Minor considerations :
1) The manuscript is well written, however as minor comment I recommend doing a revision of the English language to avoid the repetition of terms to describe the function, or genetic levels, such as "increased" or "decreased", and use / combine more as "triggers" or "promotes". However, I reiterate that the grammatical level of the text is adequate.
2) I recommend rewriting the second half of the abstract to further emphasize the aim and the conclusions, for future readers
3) The introduction is appropiated but I suggest the authors rewrite sentences 70 to 91 making the aim clearer and more direct, while the last sentence is much more conclusive about the work developed and the results presented
4) Methods 2.7 to 2.10 should be rewritten in order to understand better the procedure followed when collecting and analyzing the data
Major considerations :
1) Regarding figure 2, I suggest: Is there an effect of the different compounds on ATF3 at different times? It is more logical to first do the dose response and time course and then establish the comparison with the other compounds.
2) Figure 3 : should be completed. Have the authors detected lipid mobilization through lipid droplets? for example using the BODIPY or SUDAN III probe?
3) In Figure 3: I recommend analyzing the contribution of beta-oxidation of fatty acids as a control of lipid mobilization. Phospholipase A? or even the use of controls with Oleic Acid. Authors know the role of AMPk/mTORC1 in their cell model?
Moderate editing of English language
Intro and Abstract should be rewritten
Author Response

(The authors gave the same response as above.)

Reviewer 3 Report
The manuscript submitted to Biomedicines discusses the benefits of a new compound with possible future implications in obesity treatment. The Introduction offers a clear picture of the current literature. The methods are adequately described. the Results are presented clearly, including explained figures. The references are well-selected. The conclusions are supported by the data.
However, the following comments should be addressed:
1. The title contains too many characters. Therefore it is difficult to read and understand.
2. Keywords could be revised according to MeSH on Demand. For unknown terms, their full name should be displayed followed by a short name.
Author Response

(The authors gave the same response as above.)

Round 2
Reviewer 1 Report
Thank you for taking my comments/suggestions into consideration. I have nothing further.
Reviewer 2 Report
Dear authors
thanks so much for your reply
Best
Dear Authors
english lenguage was improved
Best